# Feasibility and Safety of an Outdoor-Simulated Interactive Indoor Cycling Device for Cardiac Rehabilitation: A Pilot Validation Study

**DOI:** 10.3390/jcm14248947

**Published:** 2025-12-18

**Authors:** Jin Taek Lee, Bo Ryun Kim, Sung Bom Pyun, Young Mo Kim, Ho Sung Son, Jae Seung Jung, Hee Jung Kim

**Affiliations:** 1Department of Physical Medicine and Rehabilitation, Korea University Anam Hospital, Seoul 02841, Republic of Korea; mutaek96@gmail.com (J.T.L.); rmpyun@korea.ac.kr (S.B.P.); rmkimym@gmail.com (Y.M.K.); 2Department of Thoracic and Cardiovascular Surgery, Korea University Anam Hospital, Seoul 02841, Republic of Korea; hssonmd@korea.ac.kr (H.S.S.); heartistcs@korea.ac.kr (J.S.J.); heejung440@hanmail.net (H.J.K.)

**Keywords:** cardiac rehabilitation, cardiovascular disease, outdoor-simulated cycling, exercise intensity, exercise test, safety

## Abstract

**Background/Objectives**: To examine the safety and clinical applicability of a newly developed outdoor-simulated interactive indoor cycling device as a potential exercise modality for cardiac rehabilitation (CR) in patients with cardiovascular disease (CVD). **Methods**: Twenty patients with CVD with low-to-moderate cardiovascular exercise risk performed a symptom-limited cardiopulmonary exercise stress test (CPET) using a modified Bruce protocol to obtain peak cardiopulmonary responses. After a 30–60 min rest, they rode the outdoor-simulated interactive indoor cycling device for 10 min with continuous gas exchange and electrocardiography monitoring. The treadmill-based CPET results were compared with those from the cycling device, focusing on key cardiopulmonary variables, such as VO_2_, HR, METs, and Rating of Perceived Exertion (RPE). **Results**: The 20 male participants had a mean age of 56.1 years. When treadmill peak values were used as reference, the cycling device elicited responses corresponding to moderate-to-vigorous intensity. In subgroup analysis, treadmill-derived peak VO_2_, peak HR, and peak MET values were significantly lower in patients aged ≥60 years compared with those aged <60 years. However, no significant differences were observed in cycling-derived values between the two groups, suggesting that cycling may represent a relatively higher-intensity exercise compared with treadmill in older patients. No significant adverse cardiac events were observed during cycling. **Conclusions**: The outdoor-simulated interactive indoor cycling device delivered exercise intensity within the therapeutic range recommended for CR in patients with CVD. Furthermore, it appeared to elicit relatively higher exercise intensity in older patients, supporting its potential as a safe and effective alternative exercise modality for CR.

## 1. Introduction

Cardiovascular disease (CVD) remains a leading cause of morbidity and mortality worldwide, often resulting in recurrent events and reduced quality of life [1]. Cardiac rehabilitation (CR) is an established, evidence-based intervention that improves prognosis, reduces hospitalizations, and enhances quality of life in patients with coronary heart disease and other forms of CVD [2,3,4]. Aerobic exercise is the cornerstone of CR because regular training improves cardiorespiratory fitness, functional capacity, and multiple cardiovascular risk factors [2,3]. In clinical practice, the treadmill and lower-limb ergometers are the most commonly used devices for supervised aerobic training because they are practical in indoor clinical settings, allow exercise regardless of weather, and provide reliable, reproducible workloads for cardiopulmonary testing and training [5].

However, strictly, indoor exercise modalities do not replicate the sensory, environmental, and motivational elements associated with outdoor activity. Exercise performed in natural or outdoor settings has been linked to greater enjoyment, intrinsic motivation, and intentions to repeat activity compared with equivalent indoor exercise, although measured physiological benefits are modest and vary across studies [6,7]. Cycling is a widely used and sustainable form of aerobic activity in daily life and offers an ecologically valid mode for assessing and prescribing exercise [8,9]. For older adults and patients with CVD, however, outdoor cycling raises safety concerns, such as falls and traffic-related injuries, along with environmental unpredictability and challenges in individualizing exercise intensity. These factors limit its routine application in supervised CR programs [10,11].

To bridge this gap, several indoor technologies, such as immersive and non-immersive virtual reality (VR) exergaming systems, integrated smart-trainer platforms, and bicycle simulators, have been developed to reproduce outdoor exercise experiences while maintaining clinical safety and workload control [7,12]. These systems can increase patient engagement and appear feasible as adjuncts to CR, although physiological responses vary by device and population [12,13,14]. Unlike earlier VR-based or stationary cycling systems, the Ulti-racer series—the outdoor-simulated interactive indoor cycling device assessed in this study—was designed to reproduce key sensory and mechanical aspects of outdoor cycling (visual flow, lateral sliding and tilting, and realistic posture dynamics) while allowing continuous electrocardiography (ECG) and breath-by-breath physiological monitoring, individualized workload control, and supervised safety measures.

In this study, we aimed to evaluate the feasibility, safety, and ability of this outdoor-simulated interactive indoor cycling device to elicit exercise intensities within established CR therapeutic ranges in patients with CVD.

## 2. Materials and Methods

### 2.1. Study Population

In this single-center prospective study, we enrolled clinically stable patients with CVD who were referred to the CR program at Korea University Anam Hospital. The study protocol was approved by the Institutional Review Board of Korea University Anam Hospital (approval no.: No. 2023AN0287). Patients were recruited consecutively between September 2023 and December 2023. During this period, only individuals participating in clinically stable phase III (maintenance stage) CR were considered for enrollment, and all eligible outpatients who met the predefined inclusion criteria were approached. Patients undergoing phase II (recovery stage) CR, as well as those who were clinically unstable or decompensated, were excluded for safety and protocol consistency. Among eligible candidates, those who were able to perform cycling exercise and provided written informed consent were enrolled on a first-come, first-served basis, resulting in a total of 20 participants.

### 2.2. Inclusion Criteria

Participants were eligible for enrollment if they met all of the following criteria:Age ≥ 18 years at enrollment.Diagnosis of CVD, including acute coronary syndrome or stable ischemic heart disease after percutaneous coronary intervention or coronary artery bypass grafting; valvular heart disease with valve replacement surgery; stable heart failure; or prior cardiac surgery.Classified as low-to-moderate exercise risk for supervised exercise and exercise testing based on institutional assessment.Ability to understand study procedures and provide written informed consent.Physical ability to mount and pedal a cycle and perform symptom-limited cardiopulmonary exercise testing (CPET).

### 2.3. Exclusion Criteria

Participants were excluded if any of the following applied:Absolute contraindications to exercise testing or training per established exercise testing guidelines [15].Hemodynamically unstable conditions (e.g., acute infection or severe electrolyte imbalance) or uncontrolled hypertension (systolic blood pressure ≥180 mmHg or diastolic blood pressure ≥110 mmHg at rest that could not be safely controlled before testing).Severe or uncontrolled arrhythmia or the presence of an implantable cardioverter defibrillator (ICD) for whom device management could not ensure safety during supervised exercise.Recent acute myocardial infarction (MI), unstable coronary syndrome, or major cardiac procedure within the acute post-event period (patients within the first 2 weeks after an acute MI were excluded).Left ventricular ejection fraction (LVEF) <30% or New York Heart Association (NYHA) class IV symptoms.Severe cognitive impairment or psychiatric disorder that, in the investigator’s judgment, prevented informed consent or safe participation.Significant musculoskeletal, neurologic, or other lower-extremity conditions that prevent safe cycling or valid CPETInability to ambulate or transfer safely for cycle mounting or dismounting without reasonable assistanceOther comorbid conditions expected to significantly limit life expectancy or interfere with participation, as judged by the investigator.

Demographic and clinical characteristics are summarized in Table 1.

### 2.4. Study Protocol

Baseline assessments were performed before any exercise testing and included body mass index (BMI), skeletal muscle index (SMI), Korean Activity Scale/Index (KASI), EuroQol-5 Dimension (EQ-5D), handgrip strength, and a 6 min walk test (6MWT). All participants first completed symptom-limited CPET on a treadmill using a modified Bruce protocol to ensure uniformity and reduce test–retest variability. During the test, breath-by-breath gas analysis was performed using a portable metabolic system (K5, COSMED Inc., Rome, Italy), and continuous ECG monitoring was performed using a wearable ECG patch device (HiCardi, MEZOO Co., Ltd., Wonju, Republic of Korea). Device-specific calibration (gas analyzer calibration and flow calibration) was performed before testing, and mask fitting was performed for each participant; the fit was checked, and any leaks were corrected until an adequate seal was confirmed. The primary purpose of the treadmill CPET was to determine each participant’s peak cardiopulmonary response, such as peak oxygen consumption (VO_2_) and peak metabolic equivalents (METs), which served as the individualized reference for subsequent comparisons.

After completing CPET and a supervised recovery period of 30–60 min, each participant was prepared for the cycling session using the outdoor-simulated interactive indoor cycling device (Ulti-racer P, Real Design Tech Co., Ltd., Seongnam, Republic of Korea). The 30–60 min rest interval was designed to minimize residual fatigue. A standardized seat and pedal fitting procedure was performed (saddle height adjusted to achieve approximately 5–10° knee flexion during the downstroke, pedal straps secured), and bicycle settings (tire/air pressure or trainer resistance) were adjusted to each participant’s anthropometry and comfort. Participants completed a brief familiarization period (typically < 1 min) to confirm comfort, seating position, and pedal engagement. Participants then performed a 10 min exercise bout on the cycling device. Participants were instructed to hold the handlebars with both hands and pedal at a self-selected cadence and power level; duration was fixed at 10 min to evaluate steady-state cardiopulmonary responses under realistic self-paced conditions. The same physiological variables measured during CPET were recorded during the cycling session. The primary purpose of the cycling session was to quantify cardiopulmonary responses produced by the outdoor-simulated cycling device and compare them with the exercise intensity range recommended for CR (40–85% of treadmill-derived peak METs obtained from the treadmill-based CPET).

To ensure participant safety and enable rapid responses to unexpected events (e.g., falls, chest pain, syncope), a physician, nurse, and physical therapist were present and continuously monitored all tests. Test termination criteria followed established CPET and CR guidelines and were documented in the study protocol. All emergency equipment, including a defibrillator, was immediately available.

Furthermore, to evaluate age-related differences, subgroup analyses were conducted by dividing patients into younger (<60 years) and older (≥60 years) groups. The aim was to determine whether the outdoor-simulated indoor cycling device could deliver exercise intensities appropriate for CR in both subgroups.

### 2.5. An Outdoor-Simulated Interactive Indoor Cycling Device

We employed a novel outdoor-simulated interactive indoor cycling device (Ulti-racer P). The system consists of four main components: a base frame, anterior and posterior rollers, a sliding module, and an adjustable mounting clamp that attaches securely to the bicycle frame. This patented frame-clamp mechanism stabilizes the bicycle on the platform while allowing the wheels to rotate freely on the rollers, thereby enabling safe and continuous indoor pedaling (Figure 1 and Figure 2) [16].

A distinctive feature of this platform is its ability to reproduce coronal-plane motion through both lateral sliding (±10 cm) and tilting (±1°). These dynamic capabilities mimic outdoor cycling balance demands while maintaining a controlled and safe indoor environment. Consequently, users can experience realistic cycling motion without the risk of falling, making the device suitable for frail older adults or patients with balance limitations, as well as healthy individuals. Importantly, the system can be operated without additional pedal sensors, ensuring versatility for research and clinical rehabilitation applications.

### 2.6. Outcome Measurements

#### 2.6.1. Assessment of Cardiopulmonary Responses

The primary variables included peak VO_2_, peak ventilatory threshold (VT), peak METs, peak heart rate (HR), resting HR, peak respiratory exchange ratio (RER), peak rate–pressure product (RPP), peak rating of perceived exertion (RPE), peak systolic and diastolic blood pressure (BP), and total exercise duration. Peak VO_2_ was defined as the highest 30 s averaged value during the test and was also expressed in METs (1 MET = 3.5 mL O_2_·kg^−1^·min^−1^) to indicate exercise intensity. Peak HR was defined as the mean HR recorded during the last 30 s of exercise and was also expressed as a percentage of the age-predicted maximal HR using the formula [HR peak/(220 − age)] × 100. RER was calculated as the ratio of $V^{\cdot }{\mathrm{CO}}_{2}$ to $V^{\cdot }O_{2}$. We considered a peak RER ≥ 1.10 a conservative indicator of maximal metabolic effort. Because the mean RER during cycling was 1.0 in our sample, cycling results were interpreted as representing high submaximal to near-maximal effort and were complemented with peak RPE and VO_2_ plateau assessments. RPE was obtained immediately at test termination using the Borg 6–20 scale. BP was measured at rest and at peak exercise using an automated sphygmomanometer (TANGO M2, SunTech Medical, Morrisville, NC, USA). Relative cardiopulmonary workload was quantified as the ratio of cycling-to-treadmill peak responses for each parameter (VO_2_, HR, METs), expressed as a percentage using the formula:Relative workload (%) = (cycling value/treadmill peak value) × 100.

#### 2.6.2. Assessment of Body Composition

BMI was calculated as weight in kilograms divided by height in meters squared (kg/m^2^). SMI was assessed using a multi-frequency bioelectrical impedance analyzer (InBody, InBody Co., Ltd., Seoul, Republic of Korea). SMI was calculated as appendicular skeletal muscle mass divided by height squared (kg/m^2^), a method validated as a reliable measure of muscle mass in clinical research [17].

#### 2.6.3. Assessment of Physical Activity Level

Physical activity level was evaluated using the KASI [18], a validated questionnaire specifically developed to assess the daily activity status of patients with CVD. The KASI provides a numerical score that reflects an individual’s habitual activity and exercise tolerance.

#### 2.6.4. Assessment of Health-Related Quality of Life

Health-related quality of life was assessed using the EQ-5D [19], which measures five domains: mobility, self-care, usual activities, pain/discomfort, and anxiety/depression. The validated Korean version was used, and the EQ-5D index score was calculated based on Korean population preference weights.

#### 2.6.5. Assessment of Muscle Strength

Handgrip strength was measured using a digital hand dynamometer (JAMAR PLUS+ Digital Hand Dynamometer; Sammons Preston Rolyan, Bolingbrook, IL, USA). Participants were instructed to stand with the arm at the side and elbow extended and then to exert maximal force for 3 s. Each hand was tested twice, with at least 1 min of rest between trials, and the highest value for each hand was recorded in kilograms for analysis [20]. Handgrip strength provides a surrogate measure of overall muscular fitness and functional capacity in patients undergoing CR [21].

#### 2.6.6. Assessment of Exercise Capacity

Exercise capacity was assessed using the 6MWT, following standardized guidelines [22,23]. Participants were instructed to walk back and forth along a flat, straight, 30 m corridor for 6 min at their maximal self-selected pace. Standardized encouragement was given at regular intervals, and the total distance walked (m) during the 6 min period was recorded as the outcome measure.

#### 2.6.7. Patient Satisfaction Assessment with Outdoor-Simulated Indoor Cycling Device

At the end of the study period, participants completed a self-administered questionnaire assessing satisfaction with the outdoor-simulated indoor cycling device. Satisfaction was rated on a 5-point Likert scale (1 = very dissatisfied to 5 = very satisfied). Participants also indicated reasons for their ratings using predefined options and were given the opportunity to provide open-ended comments regarding specific advantages or areas for improvement.

### 2.7. Safety Monitoring

Continuous single-lead ECG monitoring was performed during the entire session using the Hicardi system (MEZOO Co., Ltd., Wonju, Republic of Korea), a wireless ECG recorder. The primary purpose was to detect potential arrhythmias and ensure patient safety during exercise. Specific arrhythmias of interest included ventricular tachycardia, ventricular fibrillation, atrial fibrillation, and advanced atrioventricular block. In addition, the occurrence of premature atrial or ventricular contractions, as well as episodes of sinus tachycardia, was recorded and analyzed for clinical relevance.

### 2.8. Statistical Analysis

All statistical analyses were performed using IBM SPSS Statistics for Windows, version 23.0 (IBM Corp., Armonk, NY, USA). Continuous variables, including demographic characteristics (age, height, weight, BMI), SMI, KASI, EQ-5D, grip strength (best value), and 6 min walk distance (6MWD), were compared between age-stratified subgroups (<60 vs. ≥60 years). Normality for each variable within each subgroup was assessed using the Shapiro–Wilk test. When both subgroups satisfied the assumption of normality (*p* ≥ 0.05), independent samples *t*-tests were performed; otherwise, Mann–Whitney U tests were applied. CPET parameters from both treadmill- and cycle-derived assessments were analyzed similarly. Subgroup comparisons (<60 vs. ≥60 years) for each modality followed the same procedure (Shapiro–Wilk test followed by either independent samples *t*-test or Mann–Whitney U test, as appropriate). Within-group comparisons between treadmill- and cycle-derived CPET results were also performed. For this analysis, individual difference scores (treadmill-cycle) were calculated for each participant. Normality of these difference scores was evaluated using the Shapiro–Wilk test. When normally distributed, paired samples t-tests were used; otherwise, Wilcoxon signed-rank tests were applied. A two-tailed *p*-value <0.05 was considered statistically significant. Results are presented as mean ± standard deviation (SD) for normally distributed data or as median (interquartile range [IQR]) for non-normally distributed data.

## 3. Results

### 3.1. Demographic Characteristics and Baseline Assessment of Participants

A total of 20 patients were included in the analysis, with a mean age of 56.1 ± 11.7 years. When stratified by age, the subgroup younger than 60 years (*n* = 11) had a mean age of 47.5 ± 7.5 years, whereas those aged 60 years or older (*n* = 9) had a mean age of 66.6 ± 5.2 years (*p* < 0.001). Patients in the younger subgroup were significantly taller (177.4 ± 8.2 cm vs. 170.6 ± 4.3 cm, *p* = 0.038) and heavier (81.3 ± 12.0 kg vs. 70.4 ± 8.5 kg, *p* = 0.035) than those in the older subgroup. BMI and SMI did not differ significantly between groups (BMI: 25.8 ± 3.0 vs. 24.2 ± 2.2, *p* = 0.210; SMI: 8.5 ± 0.5 vs. 8.0 ± 0.6, *p* = 0.099).

Grip strength was significantly higher in the younger subgroup (43.2 ± 4.1 kg vs. 38.0 ± 5.8 kg, *p* = 0.029). In contrast, KASI and EQ-5D did not follow normal distributions; therefore, values are reported as median (IQR:KASI: 67.8 [IQR = 19.8] vs. 71.4 [IQR = 20.5], *p* = 0.824; EQ-5D: 0.95 [IQR = 0.0] vs. 0.95 [IQR = 0.0], *p* = 0.766). The 6MWT distance did not differ between the subgroups (557.7 ± 62.7 m vs. 565.3 ± 76.4 m, *p* = 0.809). Baseline characteristics stratified by age (<60 vs. ≥60 years) are summarized in Table 1.

### 3.2. Comparison of Cardiopulmonary Responses Between the Treadmill and the Cycling Device

During treadmill-based CPET, peak values were as follows: VO_2_ (28.4 ± 5.6 mL·kg^−1^·min^−1^), HR (152.8 ± 19.8 bpm), METs (8.2 ± 1.6), RER (1.1 ± 0.1), and RPE (14.6 ± 1.1). With the outdoor-simulated interactive indoor cycling device, the corresponding mean values were VO_2_ (19.8 ± 5.7 mL·kg^−1^·min^−1^), HR (134.9 ± 18.4 bpm), METs (5.6 ± 1.4), RER (1.0 ± 0.1), and RPE (13.2 ± 1.7). Cycling elicited 69.7% of treadmill-derived peak VO_2_, 68.3% of the treadmill-derived peak METs, and 88.3% of treadmill-derived peak HR (Table 2 and Table 3).

According to the American College of Sports Medicine (ACSM) guidelines [24,25], the cycling-induced VO_2_ value (19.8 mL·kg^−1^·min^−1^; 69.7% of treadmill-derived peak VO_2_) corresponded to a vigorous-intensity range (64–90% VO_2_ max). The MET value (5.6 METs; 68.3% of treadmill-derived peak METs) also fell within the vigorous-intensity range when expressed as a percentage of maximal METs. HR reached 88.3% of treadmill-derived peak HR, consistent with vigorous intensity. The mean perceived exertion during cycling was 13.2 ± 1.7 on the Borg 6–20 scale, indicating moderate-to-vigorous intensity (moderate: 12–13; vigorous: 14–17) (Table 4).

According to the European Association of Preventive Cardiology/European Society of Cardiology (EAPC/ESC) guidelines [25], the cycling-induced VO_2_ (19.8 mL·kg^−1^·min^−1^; 69.7% of treadmill-derived peak VO_2_) corresponds to moderate intensity (40–69% of VO_2_ max). The MET value (5.6 METs; 68.3% of treadmill-derived METs) also falls within the moderate-intensity range when expressed as a percentage of maximal METs. In contrast, the achieved HR (88.3% of treadmill-derived peak HR) reflects vigorous intensity. The mean perceived exertion during cycling was 13.2 ± 1.7 on the Borg 6–20 scale, indicating moderate-to-high intensity (moderate: 12–13; high: 14–16) (Table 4). Thus, both frameworks indicate that the novel cycling modality elicited moderate-to-vigorous exercise intensity for CR, which aligns with the aerobic training intensity recommended to improve cardiorespiratory fitness in CR.

### 3.3. Subgroup Analysis Regarding the Comparison of Cardiopulmonary Responses Between the Treadmill and the Cycling Device

On the treadmill, the younger group (<60 years) demonstrated significantly greater exercise capacity than the older group (≥60 years). Peak VO_2_ was higher in the younger group (31.8 ± 4.7 vs. 24.1 ± 3.1 mL·kg^−1^·min^−1^, *p* = 0.001), as were peak HR (163.6 ± 16.2 vs. 137.6 ± 15.4 bpm, *p* = 0.002), peak METs (9.1 ± 1.3 vs. 6.9 ± 0.9, *p* < 0.001), and peak RPE (14.8 ± 1.1 vs. 14.2 ± 1.0, *p* = 0.295). In contrast, during the cycling device test, peak VO_2_ (19.6 ± 5.0 vs. 20.0 ± 6.9 mL·kg^−1^·min^−1^, *p* = 0.897), peak HR (141.3 ± 19.1 vs. 127.7 ± 15.5 bpm, *p* = 0.102), peak METs (5.6 ± 1.4 vs. 5.7 ± 1.9, *p* = 0.904), and peak RPE (12.7 ± 1.6 vs. 13.8 ± 1.9, *p* = 0.331) were comparable between the younger (<60 years) and older groups (≥60 years), with no significant between-group differences (all *p* > 0.05) (Table 2, Table 3 and Table 4).

These findings suggest that, for older patients, cycling exercise may impose a relatively higher cardiopulmonary workload than treadmill exercise, potentially enabling more effective aerobic stimulation within a safe intensity range.

### 3.4. Adverse Events

Continuous ECG monitoring during all exercise sessions revealed no clinically significant arrhythmias; specifically, no episodes of ventricular tachycardia, ventricular fibrillation, atrial fibrillation or second- or third-degree AV block were detected. Minor, transient ectopic beats were observed in a small number of participants but were not associated with symptoms or hemodynamic instability. No falls, musculoskeletal injuries, or other exercise-related adverse events occurred during any session.

### 3.5. Patient Satisfaction with the Novel Cycling Device

Patient satisfaction with the novel cycling device was evaluated after program completion. Of the 20 participants, 19 completed the satisfaction survey. On the 5-point Likert scale, nine participants (47.4%) reported being “very satisfied,” eight (42.1%) were “satisfied,” and only one (5.3%) selected “neutral” or “dissatisfied.” The one dissatisfied participant cited pedal discomfort despite the realistic cycling experience. Participants most valued the device’s ability to support balance and whole-body training (70%), followed by its potential use in rehabilitation settings (65%), novelty and engagement (35%), and realism of outdoor-like cycling (30%). Open-ended comments emphasized: (1) reassurance from continuous cardiac monitoring, (2) enhanced balance and combined limb management, (3) enjoyment and novelty, (4) realistic cycling dynamics, and (5) convenience. Overall, both quantitative and qualitative responses showed high satisfaction and acceptance, highlighting the device’s enjoyable and rehabilitative value, with minor concerns regarding pedal comfort.

## 4. Discussion

In this pilot study of clinically stable patients with CVD, we evaluated cardiopulmonary responses and safety of an outdoor-simulated interactive indoor cycling device compared with symptom-limited treadmill CPET. The principal findings were threefold. First, the cycling device generated a significant aerobic stimulus, eliciting approximately 69.7% of treadmill-driven peak VO_2_ (19.8 vs. 28.4 mL·kg^−1^·min^−1^), 68.3% of treadmill-derived peak METs (5.6 vs. 8.2), and 88.3% of treadmill-derived peak HR (134.9 bpm vs. 152.8 bpm). In addition, perceived exertion and RER were slightly lower during cycling (RPE = 13.2 ± 1.7 vs. 14.6 ± 1.1; RER = 1.0 ± 0.1 vs. 1.1 ± 0.1), indicating that the device achieved moderate-to-vigorous intensity appropriate for CR prescription. Second, no major adverse events occurred during device use under supervised conditions, and arrhythmic or hemodynamic abnormalities were infrequent and transient. In addition, no exercise-related adverse events, such as falls or musculoskeletal injuries, were observed throughout the sessions [26]. Third, participants reported favorable acceptance and overall satisfaction with the novel cycling device.

In the context of CR, aerobic exercise remains a cornerstone intervention that improves cardiorespiratory fitness, functional capacity, and long-term prognosis in patients with CVD [4,24,27,28]. The present study demonstrated that an outdoor-simulated interactive indoor cycling device can safely provide a significant aerobic workload, aligning with ACSM and EAPC/ESC recommendations for CR training intensities. Importantly, the absence of adverse cardiovascular or musculoskeletal events indicates that this device can deliver an aerobic stimulus within a clinically safe range.

From a practical standpoint, the novel cycling device may serve as an alternative or adjunct to traditional treadmill or stationary ergometer exercise in CR programs. Given that treadmill walking may be limited by gait imbalance, joint discomfort, or fear of falling—particularly in older or deconditioned patients—this device provides a stable, immersive, and engaging environment that may enhance adherence and support safe intensity progression. Furthermore, its outdoor-stimulated riding experience may enhance patient motivation and perceived enjoyment, factors known to improve long-term exercise compliance.

In addition, these findings align with contemporary CR guidelines that emphasize individualized exercise prescription and multimodal training approaches to optimize patient engagement and outcomes [29,30].

Therefore, the outdoor-simulated cycling device could be feasibly incorporated into hybrid or center-based CR settings as a safe and effective option for eliciting moderate-to-vigorous aerobic exercise intensity, particularly for patients unable or unwilling to perform treadmill-based training. Importantly, from a practical CR perspective, the device’s ability to provide a realistic outdoor-cycling experience while delivering a clinically useful cardiovascular workload supports its potential role as an alternative or adjunct to standard modalities, particularly for patients who prefer cycling or who have gait limitations.

In our subgroup analysis, while treadmill-based CPET revealed significantly lower cardiopulmonary responses in patients aged ≥ 60 years than in those aged < 60 years, the outdoor-simulated cycling device produced no significant differences between the age strata (VO_2_ mean: 19.6 ± 5.0 vs. 20.0 ± 6.7 mL·kg^−1^·min^−1^, *p* = 0.897; METs: 5.6 ± 1.4 vs. 5.7 ± 1.9, *p* = 0.904). This finding suggests that the novel cycling device may represent a higher relative workload for older patients, who might struggle to reach sufficient intensity during walking-based exercise. Given that the device also demonstrated a favorable safety profile, with no falls or musculoskeletal injuries observed, it appears to be a stable and effective modality for aerobic training in older cardiovascular patients in CR settings. These attributes support its potential integration into CR programs, particularly for patients who may be limited in traditional treadmill-based training due to age-related decline, gait instability, or comorbid musculoskeletal conditions.

Beyond cardiovascular capacity alone, several biomechanical and perceptual mechanisms may help explain the differential responses between treadmill and cycling exercise observed in our study. Cycling imposes lower demands on postural stabilization and balance control compared with treadmill walking, which may allow older individuals to tolerate higher relative workloads [16,31,32]. Because cycling primarily engages large proximal muscle groups, such as the quadriceps and gluteal muscles, with reduced impact loading and less step-to-step variability, patients may perceive the task as more stable and predictable, facilitating greater effort without disproportionate increases in perceived exertion [33]. Additionally, cycling reduces joint loading and fear of imbalance, factors particularly relevant to older adults, which may partially account for the blunted age-related performance gap observed during cycling but not during treadmill CPET. These neuromuscular and perceptual factors collectively support the physiological plausibility of our subgroup findings.

Notably, these observations are consistent with prior reports demonstrating the value of alternative, low-impact modalities in cardiac rehabilitation [34,35]. Recent studies have shown that such modalities can improve functional capacity while enhancing safety and patient engagement, particularly in populations with balance limitations or musculoskeletal concerns [16,35]. For example, structured cycling-based or hybrid exercise programs have been successfully implemented in cardiovascular rehabilitation settings, yielding improvements in aerobic fitness, patient adherence, and overall functional recovery [36,37]. Such evidence—including emerging reports of non-weight-bearing or reduced-stabilization exercise strategies—further supports the potential role of outdoor-simulated cycling devices as accessible and safe training options within contemporary cardiac rehabilitation paradigms.

Safety is an essential consideration for any new exercise modality introduced into CR. The present study involved a single supervised session; however, continuous monitoring revealed no sustained ventricular arrhythmias, no events requiring urgent intervention, and only occasional, asymptomatic ectopy. These findings support the feasibility of supervised implementation in selected low-to-moderate-risk patients [26]. Nevertheless, generalization to higher-risk populations requires caution and further investigation.

In our study, high levels of patient satisfaction with the novel outdoor-simulated cycling device were observed, suggesting that it may enhance motivation and engagement in CR. Given that sustained exercise adherence is one of the most important determinants of long-term improvements in cardiorespiratory fitness and clinical outcomes, this finding is particularly relevant. A systematic review and meta-regression of randomized controlled trials reported that adherence to prescribed exercise sessions was the strongest predictor of improvement in peak VO_2_ and contributed significantly to reductions in cardiovascular mortality and rehospitalization [38].

Despite the well-established benefits of CR, real-world participation and long-term adherence remain suboptimal, with a meta-analysis reporting participation rates of approximately 34% among eligible patients [39]. Because the novel cycling device offers a more engaging, immersive, and potentially less intimidating alternative to traditional treadmill or stationary ergometer modalities, it may help overcome common participation barriers (such as fear of exercise, balance limitations, or boredom) and thus improve adherence. Enhanced adherence, in turn, is likely to lead to sustained gains in aerobic capacity, improved secondary prevention, and lower mortality in patients with CVD. Overall, the novel device appears to be a promising modality for supervised CR: it elicits clinically meaningful cardiovascular stress within recommended training ranges, demonstrates an acceptable safety profile in a selected population, and is perceived positively by users.

Although the present study focused on a single supervised session, the findings suggest potential pathways for future integration of this device into routine CR programming. Clinically stable patients in phase III CR may represent an appropriate initial population, with the possibility of extending its use to earlier recovery stages as evidence accumulates. To better define its long-term applicability, future trials should incorporate longer continuous exercise sessions and standardized progression schemes. Such studies would clarify how this modality can be embedded into structured CR protocols and inform its role across different phases of CR.

### Limitations

This study has some important limitations that should be acknowledged. First, it was a small, single-center pilot study with a total of 20 participants. The limited sample size reduces the statistical power and generalizability of the findings; therefore, the results should be interpreted as exploratory and hypothesis-generating rather than confirmatory. Second, all participants were male, and the absence of female participants prevents assessment of potential sex-related differences in physiological responses, limiting the external validity of the findings to male patients only. However, participants were recruited consecutively during the study period, and enrollment was based solely on predefined eligibility criteria and willingness to participate. Therefore, although the sample was limited, intentional selective recruitment was minimized. Third, there was a selection bias toward low-to-moderate-risk patients, as all participants were clinically stable and suitable for supervised exercise testing. Consequently, the results may not be applicable to higher-risk cardiovascular populations. Fourth, the study did not include an age-matched healthy control group. The inclusion of a non-cardiac control group would have allowed clearer interpretation of whether observed differences between treadmill and cycling modalities were related to cardiac pathology or the intrinsic characteristics of each exercise modality and would also have provided normative reference values for evaluating percent-of-peak responses. Fifth, all participants completed the treadmill CPET first, followed by the cycling test after a 30–60 min recovery period. This fixed sequence may have introduced order or fatigue effects, potentially contributing to lower performance on the cycling test. A randomized crossover design would have better controlled for such sequence bias. Sixth, the cycling session was limited to 10 min, which is shorter than the typical 30- to 40 min duration of standard cardiac rehabilitation exercise sessions. Although this duration was sufficient for assessing acute cardiopulmonary responses in this pilot setting, a longer continuous session would have more closely reflected real-world CR training and may have provided additional information regarding sustained physiological responses, symptom development, and practicality for session-based rehabilitation. Finally, monitoring during exercise sessions was limited to in-session physiological surveillance and short-term post-test observation. Long-term outcomes, including adherence, training-induced peak VO_2_ improvement, quality-of-life changes, and cardiovascular events, were not evaluated. Future studies with comprehensive monitoring and longitudinal follow-up are warranted to clarify the clinical implications and sustainability of the observed responses.

## 5. Conclusions

An outdoor-simulated interactive indoor cycling device elicited moderate to high cardiopulmonary workloads in clinically stable patients with CVD, producing exercise intensities that fall within typical CR training ranges while demonstrating an acceptable short-term safety profile under supervision. Given its interactive and immersive nature, this novel device may be safely applicable even in older patients with CVD, offering a practical and enjoyable indoor aerobic exercise modality that simulates outdoor cycling. Its potential to enhance patient engagement and exercise adherence suggests meaningful applicability as a complementary tool in modern CR programs.

## Figures and Tables

**Figure 1 jcm-14-08947-f001:**
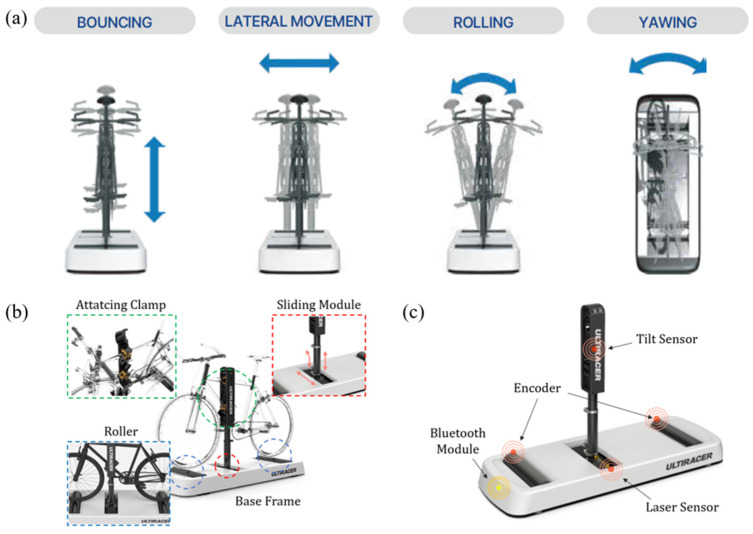
Outdoor-simulated interactive indoor cycling device (**a**) dynamics of cycling device, (**b**) structure of cycling device, (**c**) sensors equipped on the cycling platform [16].

**Figure 2 jcm-14-08947-f002:**
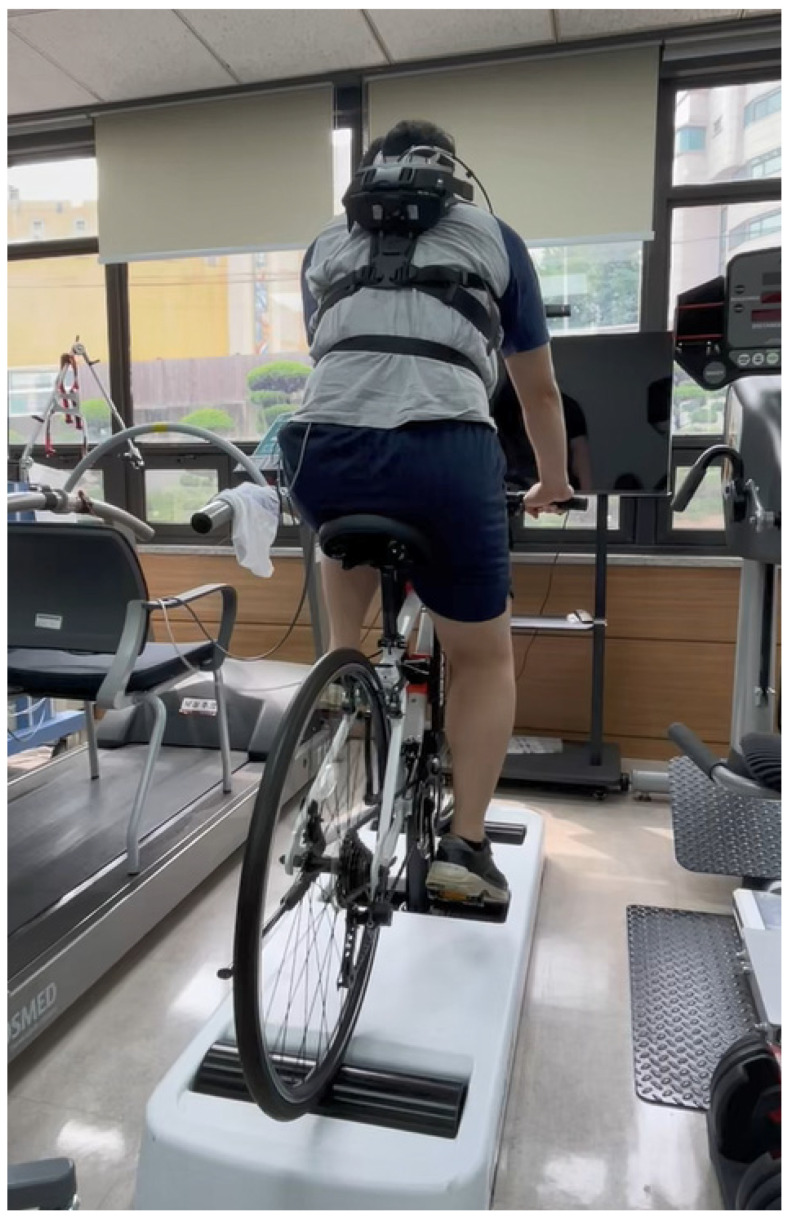
Experimental setting of the outdoor-simulated interactive indoor cycling device.

**Table 1 jcm-14-08947-t001:** Demographic characteristics and baseline clinical assessments of the participants (*N* = 20).

		Age Group		
	Total	Under 60	Over 60	*p*-Value
	(*N* = 20)	(*N* = 11)	(*N* = 9)	
Age (years)	56.1 (±11.7)	47.5 (±7.5)	66.6 (±5.2)	<0.001
Height (cm)	174.3 (±7.4)	177.4 (±8.2)	170.6 (±4.3)	0.038
Weight (kg)	76.4 (±11.7)	81.3 (±12.0)	70.4 (±8.5)	0.035
BMI (kg/m^2^)	25.1 (±2.8)	25.8 (±3.0)	24.2 (±2.2)	0.210
Diagnosis (number)				
Angina	7 (35%)	3 (15%)	4 (20%)	
NSTEMI	2 (10%)		2 (10%)	
STEMI	5 (25%)	4 (20%)	1 (5%)	
Heart failure	1 (5%)	1 (5%)		
Valvular heart disease	2 (10%)	1 (5%)	1 (5%)	
Aortic disease	1 (5%)		1 (5%)	
Infective endocarditis	1 (5%)	1 (5%)		
Cardiomyopathy	1 (5%)	1 (5%)		
SMI (kg/m^2^)	8.3 (±0.6)	8.5 (±0.5)	8.0 (±0.6)	0.099
KASI	66.2 (9.9)	65.7 (9.6)	66.9 (10.9)	0.824
EQ-5D	0.9 (0.0)	0.9 (0.0)	0.9 (0.0)	0.766
Hand grip strength (kg)	40.9 (±5.4)	43.2 (±4.1)	38.0 (±5.8)	0.029
6 min walk test (m)	561.2 (±67.4)	557.7 (±62.7)	565.3 (±76.4)	0.809

Values are presented as mean ± standard deviation, median (interquartile range), or number (%) of cases. BMI, body mass index; NSTEMI, non-ST-segment elevation myocardial infarction; STEMI, ST-segment elevation myocardial infarction; SMI, skeletal muscle mass index; KASI, Korean activity scale index; EQ-5D, EuroQol-5 Dimension.

**Table 2 jcm-14-08947-t002:** Cardiopulmonary exercise responses during the treadmill exercise according to age group (*N* = 20).

		Age Group		
	Total	Under 60	Over 60	*p*-Value
	(*N* = 20)	(*N* = 11)	(*N* = 9)	
Peak VO_2_ (mL·kg^−1^·min^−1^)	28.4 (±5.6)	31.8 (±4.7)	24.1 (±3.1)	<0.001
Peak VT (L/min)	15.8 (±3.9)	15.9 (±3.5)	15.7 (±4.5)	0.925
Peak HR (bpm)	151.9 (±20.3)	163.6 (±16.2)	137.6 (±15.4)	0.002
Rest HR (bpm)	75.0 (18.3)	77.0 (18.0)	70.0 (16.0)	0.295
Peak predicted HR (%)	92.1 (±9.2)	94.5 (±8.9)	89.1 (±9.3)	0.207
Peak METs	8.1 (±1.6)	9.1 (±1.3)	6.9 (±0.9)	<0.001
VE/VCO_2_	25.5 (±3.9)	25.0 (±4.1)	26.1 (±3.8)	0.531
Peak SBP (mmHg)	185.5 (±29.8)	193.6 (±28.7)	175.4 (±29.5)	0.181
Peak DBP (mmHg)	80.7 (±14.0)	85.1 (±13.4)	75.3 (±13.5)	0.124
Rest SBP (mmHg)	128.5 (±16.7)	125.8 (±16.1)	131.7 (±17.8)	0.451
Rest DBP (mmHg)	80.1 (±10.3)	84.7 (±9.1)	74.4 (±9.1)	0.022
Exercise duration (s)	902.3 (±92.8)	909.4 (±73.1)	893.7 (±116.8)	0.718
Peak RER	1.1 (±0.1)	1.2 (±0.1)	1.1 (±0.1)	0.050
Peak RPP (mmHg · bpm)	25,451.4 (±6402.8)	28,420.7 (±5699.4)	21,822.2 (±5460.8)	0.017
Peak RPE	14.6 (±1.1)	14.8 (±1.1)	14.2 (±1.0)	0.295

Values are presented as mean ± standard deviation, median (interquartile range) or number (%) of cases. VO_2_, oxygen consumption; VT, ventilatory threshold; HR, heart rate; VE/VCO_2_, ventilatory equivalent for carbon dioxide; SBP, systolic blood pressure; DBP, diastolic blood pressure; METs, metabolic equivalents; RER, respiratory exchange ratio; RPP, rate–pressure product; RPE, rating of perceived exertion. Bonferroni-adjusted significance level (adjusted alpha *): α* = 0.05/15 = 0.0033.

**Table 3 jcm-14-08947-t003:** Cardiopulmonary exercise responses during the cycling device according to age group (N = 20).

		Age Group		
	Total	Under 60	Over 60	*p*-Value
	(N = 20)	(N = 11)	(N = 9)	
Mean VO_2_ (mL·kg^−1^·min^−1^)	19.8 (±5.7)	19.6 (±5.0)	20.0 (±6.7)	0.897
Mean HR (bpm)	135.2 (±18.5)	141.3 (±19.1)	127.7 (±15.5)	0.102
Rest HR (bpm)	85.6 (±13.7)	87.5 (±12.9)	83.2 (±15.1)	0.507
Mean Predicted HR (%)	80.9 (±11.0)	79.5 (±12.4)	83.2 (±15.1)	0.546
Mean METs	5.7 (±1.6)	5.6 (±1.4)	5.7 (±1.9)	0.904
Mean VE/VCO_2_	28.7 (±7.2)	29.8 (±4.6)	27.3 (±9.6)	0.450
Mean SBP (mmHg)	174.2 (±25.1)	171.2 (±26.4)	177.8 (±24.3)	0.572
Mean DBP (mmHg)	82.2 (±11.2)	80.5 (±10.8)	84.2 (±12.0)	0.469
Rest SBP (mmHg)	119.2 (±14.6)	119.0 (±15.1)	119.3 (±14.8)	0.961
Rest DBP (mmHg)	77.6 (±11.0)	79.8 (±10.9)	74.8 (±11.2)	0.322
Exercise duration	601.0 (1.0)	601.0 (1.0)	601.0 (1.0)	0.656
Mean RER	1.0 (±0.1)	1.0 (±0.1)	0.9 (±0.1)	0.485
Mean RPP (mmHg · bpm)	22,532.5 (±4496.4)	23,083.4 (±5529.6)	21,859.1 (±2978.2)	0.559
Mean RPE	13.2 (±1.7)	12.7 (±1.6)	13.8 (±1.9)	0.331

Values are presented as mean ± standard deviation, median (interquartile range) or number (%) of cases. VO_2_, oxygen consumption; VT, ventilatory threshold; HR, heart rate; VE/VCO_2_, ventilatory equivalent for carbon dioxide; SBP, systolic blood pressure; DBP, diastolic blood pressure; METs, metabolic equivalents; RER, respiratory exchange ratio; RPP, rate–pressure product; RPE, rating of perceived exertion. Bonferroni-adjusted significance level (adjusted alpha *): α* = 0.05/14 = 0.0036.

**Table 4 jcm-14-08947-t004:** Classification of exercise intensity according to ACSM and EAPC/ESC guidelines based on measured cardiopulmonary responses during the treadmill and the cycling device.

	Treadmill	Cycling Device	Cycling/ Treadmill	ACSM-Recommended Exercise Intensity ^1^	EAPC/ESC-Recommended Exercise Intensity ^2^
VO_2_ (mL·kg^−1^·min^−1^)	28.4 ± 5.6	19.8 ± 5.7	69.7%	Moderate (46–63%) Vigorous (64–90%)	Moderate (40–69%) Vigorous (70–85%)
HR (bpm)	152.8 ± 19.8	134.9 ± 18.4	88.3%	Moderate (64–76%) Vigorous (77–95%)	Moderate (55–74%) Vigorous (75–90%)
METs	8.2 ± 1.6	5.6 ± 1.4	68.3%	Moderate (46–63%) Vigorous (64–90%)	Moderate (46–63%) Vigorous (64–90%)
RPE	14.6 ± 1.1	13.2 ± 1.7		Moderate 12–13 Vigorous 14–17	Moderate 12–13 Vigorous 14–16

Values are presented as mean ± standard deviation or percentage (%), as appropriate. ^1^ Guidelines from the American College of Sports Medicine (ACSM). ^2^ Guidelines from European Association of Preventive Cardiology (EAPC/ESC). VO_2_, oxygen consumption; HR, heart rate; METs, metabolic equivalents; RER, respiratory exchange ratio; RPE, rating of perceived exertion.

## Data Availability

The raw data supporting the conclusions of this article will be made available by the authors on request.

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
