# Peer review of "Feasibility and Safety of an Outdoor-Simulated Interactive Indoor Cycling Device for Cardiac Rehabilitation: A Pilot Validation Study"

_jcm, 2025, doi:10.3390/jcm14248947_

Round 1

Reviewer 1 Report

Comments and Suggestions for Authors

Congratulations to the authors for their really interesting manuscript; authors should correct some flaws that I’ve found in their manuscript:

The inclusion and exclusion criteria are clearly defined and appropriate to ensure participant safety; however, the recruitment approach described as “clinically stable patients referred to cardiac rehabilitation” implies the use of a convenience sample rather than consecutive enrollment. This raises the possibility of selection bias, which the authors should address and justify explicitly in the text.

The 10-minute, self-paced cycling session appears too brief to represent a standard cardiac rehabilitation exercise protocol, which typically involves 30–45 minutes of training at a prescribed intensity. The authors are encouraged to clarify the rationale behind this chosen duration or provide supporting evidence for its validity in this context.

The statistical analysis employs multiple independent t-tests without adjustment for multiple comparisons; indeed a Bonferroni correction or false discovery rate  adjustment should be applied to avoid error of type I.

the manuscript lacks a physiological explanation for the observed differences between treadmill and cycling performance. Discussion should be expanded to include relevant mechanisms, such as variations in muscle recruitment, postural stabilization demands, or sensory-motor feedback, that may underlie the differential responses between exercise modalities. Moreover authors are encouraged to include other successful examples of rehabilitation in cardiology (doi: 10.23736/S2724-5683.25.06885-1)

Author Response

First of all, we sincerely thank you for your careful review of our manuscript. We have revised it to reflect the comments you have pointed out as much as possible. The revisions are as follows.

Comments and suggestion for Authors

Congratulations to the authors for their really interesting manuscript; the authors should correct some flaws that I’ve found in their manuscript:

  1. The inclusion and exclusion criteria are clearly defined and appropriate to ensure participant safety; however, the recruitment approach described as “clinically stable patients referred to cardiac rehabilitation” implies the use of a convenience sample rather than consecutive enrollment. This raises the possibility of selection bias, which the authors should address and justify explicitly in the text.

→ We appreciate your critical and thoughtful comment. We would like to clarify that no selective screening or preferential enrollment was performed in this study. As described in the revised Methods section, Patients were recruited consecutively between September 2023 and December 2023. During this period, only individuals participating in clinically stable phase III (maintenance stage) CR were considered for enrollment, and all eligible outpatients who met the predefined inclusion criteria were approached. Patients undergoing phase II (recovery stage) CR, as well as those who were clinically unstable or decompensated, were excluded for safety and protocol consistency. Among eligible candidates, those who were able to perform cycling exercise and provided written informed consent were enrolled on a first-come, first-served basis, resulting in a total of 20 participants. Considering your critical and thoughtful comment, we have added an explanation in the method and limitation section. Please refer to lines 80-88 on page 3.

  1. The 10-minute, self-paced cycling session appears too brief to represent a standard cardiac rehabilitation exercise protocol, which typically involves 30–45 minutes of training at a prescribed intensity. The authors are encouraged to clarify the rationale behind this chosen duration or provide supporting evidence for its validity in this context.

→ We appreciate your insightful comment. Before initiating the study, two investigators conducted pilot testing of the cycling device with simultaneous gas analysis. During their preliminary trials, we observed that when cycling at a self-selected pace, oxygen uptake and other gas-exchange variables reached a stable range early in the session and remained consistent throughout a 10-minute period. These findings suggested that a 10-minute duration was sufficient to obtain representative steady-state cardiopulmonary responses for comparison with CPET-derived values. Additionally, symptom-limited CPET in the outpatient population typically lasted approximately 14-15 minutes, and our intention was not to replicate a full cardiac rehabilitation training session but to compare the exercise intensity elicited by the device relative to each patient’s CPET responses. Given these considerations, we determined that a 10-minute, self-paced bout was appropriate and sufficient for the study’s physiological aims while minimizing participant burden. Considering your critical and thoughtful comment, we have added an explanation in the limitation section. Please refer to lines 520-525 on page 16.

  1. The statistical analysis employs multiple independent t-tests without adjustment for multiple comparisons; indeed, a Bonferroni correction or false discovery rate adjustment should be applied to avoid error of type I.

→ Thank you for raising this important point regarding the potential need for multiple-comparison correction. After discussion with a biostatistics professor, Tables 2 and 3 did not include simultaneous comparisons of multiple variables because they analyzed different outcome variables rather than a single dependent variable. Therefore, we believe that there is no need to consider the error of inflating the Type I error because the outcome variables are all different. However, in consideration of your comments, we have conservatively adopted the Bonferroni-corrected significance level for Tables 2 and 3. This modified criterion (alpha*) has been explicitly noted at the foot of the relevant tables. (α / number of comparisons). Importantly, key clinical variables such as peak VOâ‚‚, peak METs, and peak HR remain significant even under this conservative value, and therefore, the main interpretation of our findings remains unchanged. We appreciate your thoughtful comment and believe that the clarification improves the clarity of our statistical reporting.

  1. The manuscript lacks a physiological explanation for the observed differences between treadmill and cycling performance. Discussion should be expanded to include relevant mechanisms, such as variations in muscle recruitment, postural stabilization demands, or sensory-motor feedback, that may underlie the differential responses between exercise modalities. Moreover, authors are encouraged to include other successful examples of rehabilitation in cardiology (doi: 10.23736/S2724-5683.25.06885-1)
    → We appreciate your critical and thoughtful comment. In response, we have expanded the discussion to include potential physiological, biomechanical, and perceptual mechanisms underlying the observed differences between treadmill and cycling exercise. Specifically, we discussed variations in muscle recruitment, postural stabilization demands, joint loading, and sensory-motor feedback, which may explain why older participants achieved comparable cycling performance despite lower treadmill responses. We also cited prior studies demonstrating the effectiveness of cycling-based or alternative modalities in cardiac rehabilitation, highlighting their safety and favorable patient acceptance. Please refer to lines 439-462 on page 14.

Reviewer 2 Report

Comments and Suggestions for Authors

The manuscript reports a single-center prospective pilot study evaluating the physiological responses, feasibility, and short-term safety of a newly developed outdoor-simulated interactive indoor cycling device in 20 clinically stable male patients with cardiovascular disease. Participants first underwent symptom-limited treadmill CPET, followed by a 10-minute session on the cycling device with continuous physiological monitoring. Cycling elicited moderate-to-vigorous intensity exercise corresponding to approximately 70% of treadmill peak VOâ‚‚, without major adverse events. Older patients showed lower treadmill performance but exhibited comparable cycling responses to younger patients, suggesting the device may provide a relatively higher workload in older adults. Patients reported high satisfaction. The authors conclude that the device is safe, feasible, and potentially useful for cardiac rehabilitation.

The study addresses an interesting and increasingly relevant topic in contemporary cardiac rehabilitation, namely the integration of immersive or outdoor-simulated cycling technologies. The manuscript is clearly structured, and the methodological description is sufficiently detailed to allow replication. The comparison with treadmill-based CPET provides an informative framework for interpreting the physiological demands of the device, and the inclusion of age-stratified analysis adds a useful clinical perspective.

A few points may help to further strengthen the work:

  1. Study design considerations: Because the treadmill test always preceded the cycling session, residual fatigue may have modestly influenced the cycling workload. Although noted in the limitations, a brief explanation of how the 30–60-minute recovery interval was chosen would clarify the authors’ reasoning.

  2. Sample composition: The exclusively male cohort limits generalizability. A short comment on whether any barriers prevented female enrollment (e.g., referral patterns or logistical constraints) may be informative for readers.

  3. Device-related details: The manuscript would benefit from a more explicit description of the decision to standardize the cycling duration at 10 minutes. Clarifying whether a longer trial might more closely reflect CR session timing would provide practical context.

  4. Interpretation of subgroup findings: The observation that older participants reached comparable cycling responses despite lower treadmill performance is intriguing. Expanding the discussion on potential biomechanical or perceptual factors—beyond simple cardiovascular capacity—could enrich the mechanistic interpretation.

  5. Long-term applicability: Although beyond the scope of this pilot, briefly outlining how this device might be integrated into routine CR programming (e.g., progression schemes, session timing, or patient selection) would highlight its practical potential.

Overall, the manuscript presents a well-executed pilot study with clear clinical relevance. Addressing the points above could further enhance the clarity and translational value of the work.

Author Response

First of all, we sincerely thank you for your careful review of our manuscript. We have revised it to reflect your comments as much as possible. The revisions are as follows.

The manuscript reports a single-center prospective pilot study evaluating the physiological responses, feasibility, and short-term safety of a newly developed outdoor-simulated interactive indoor cycling device in 20 clinically stable male patients with cardiovascular disease. Participants first underwent symptom-limited treadmill CPET, followed by a 10-minute cycling session with continuous physiological monitoring. Cycling elicited moderate-to-vigorous intensity exercise corresponding to approximately 70% of treadmill peak VOâ‚‚, without major adverse events. Older patients showed lower treadmill performance but exhibited comparable cycling responses to younger patients, suggesting the device may provide a relatively higher workload in older adults. Patients reported high satisfaction. The authors conclude that the device is safe, feasible, and potentially useful for cardiac rehabilitation.

The study addresses an interesting and increasingly relevant topic in contemporary cardiac rehabilitation, namely the integration of immersive or outdoor-simulated cycling technologies. The manuscript is clearly structured, and the methodological description is sufficiently detailed to allow replication. The comparison with treadmill-based CPET provides an informative framework for interpreting the physiological demands of the device, and the inclusion of age-stratified analysis adds a useful clinical perspective.

A few points may help to further strengthen the work:

  1. Study design considerations: Because the treadmill test always preceded the cycling session, residual fatigue may have modestly influenced the cycling workload. Although noted in the limitations, a brief explanation of how the 30–60-minute recovery interval was chosen would clarify the authors’ reasoning.

→ We thank for your thoughtful comment. Before initiating the study, two investigators performed pilot testing in which a treadmill-based CPET was followed by a 30-minute rest period and then a session on the cycling device. During these preliminary trials, both central and peripheral fatigue were specifically assessed, and neither investigator reported any residual fatigue that affected performance on the cycling device. In the actual study, a minimum rest period of 30 minutes was applied consistently for all participants. After this initial interval, patients were asked about symptoms suggestive of central or peripheral fatigue, and, when necessary, the recovery period was extended up to 60 minutes to ensure full readiness for the subsequent cycling session. This approach allowed individualized recovery while maintaining practical feasibility in an outpatient setting.  

  1. Sample composition: The exclusively male cohort limits generalizability. A short comment on whether any barriers prevented female enrollment (e.g., referral patterns or logistical constraints) may be informative for readers.

→ We appreciate your critical and thoughtful comment. We would like to clarify that no selective screening or preferential enrollment was performed in this study. As described in the revised Methods section, Patients were recruited consecutively between September 2023 and December 2023. During this period, only individuals participating in clinically stable phase III (maintenance stage) CR were considered for enrollment, and all eligible outpatients who met the predefined inclusion criteria were approached. Patients undergoing phase II (recovery stage) CR, as well as those who were clinically unstable or decompensated, were excluded for safety and protocol consistency. Among eligible candidates, those who were able to perform cycling exercise and provided written informed consent were enrolled on a first-come, first-served basis, resulting in a total of 20 participants. Considering your critical and thoughtful comment, we have added an explanation in the method and limitation section. Please refer to lines 80-88 on page 3 and lines 505-508 on page 15.

  1. Device-related details: The manuscript would benefit from a more explicit description of the decision to standardize the cycling duration at 10 minutes. Clarifying whether a longer trial might more closely reflect CR session timing would provide practical context.

→ We appreciate your insightful and valuable comment. Before initiating the study, two investigators conducted pilot testing of the cycling device with simultaneous gas analysis. During their preliminary trials, we observed that when cycling at a self-selected pace, oxygen uptake and other gas-exchange variables reached a stable range early in the session and remained consistent throughout a 10-minute period. These findings suggested that a 10-minute duration was sufficient to obtain representative steady-state cardiopulmonary responses for comparison with CPET-derived values. Additionally, symptom-limited CPET in the outpatient population typically lasted approximately 14-15 minutes, and our intention was not to replicate a full cardiac rehabilitation training session but to compare the exercise intensity elicited by the device relative to each patient’s CPET responses. Given these considerations, we determined that a 10-minute, self-paced bout was appropriate and sufficient for the study’s physiological aims while minimizing participant burden. In retrospect, using a 30-minute duration might have been preferable, as it would better reflect the format of a standard cardiac rehabilitation session. Considering your critical and thoughtful comment, we have added an explanation in the limitation section. Please refer to lines 520-525 on page 16.

  1. Interpretation of subgroup findings: The observation that older participants reached comparable cycling responses despite lower treadmill performance is intriguing. Expanding the discussion on potential biomechanical or perceptual factors—beyond simple cardiovascular capacity—could enrich the mechanistic interpretation.

→ We appreciate your critical and thoughtful comment. In response, we have expanded the discussion to include potential physiological, biomechanical, and perceptual mechanisms underlying the observed differences between treadmill and cycling exercise. Specifically, we discussed variations in muscle recruitment, postural stabilization demands, joint loading, and sensory-motor feedback, which may explain why older participants achieved comparable cycling performance despite lower treadmill responses. We also cited prior studies demonstrating the effectiveness of cycling-based or alternative modalities in cardiac rehabilitation, highlighting their safety and favorable patient acceptance. Please refer to lines 439-462 on page 14.

  1. Long-term applicability: Although beyond the scope of this pilot, briefly outlining how this device might be integrated into routine CR programming (e.g., progression schemes, session timing, or patient selection) would highlight its practical potential.

→ We thank you for your thoughtful comment regarding long-term applicability. We have added a section to the discussion outlining potential integration of the cycling device into routine cardiac rehabilitation programming. Specifically, we described its use in clinically stable phase III CR patients, potential extension to earlier recovery stages, and the need for future trials with longer continuous sessions and structured progression schemes to evaluate long-term applicability and inform implementation across different CR phases. Please refer to lines 489-496 on page 15.

Overall, the manuscript presents a well-executed pilot study with clear clinical relevance. Addressing the points above could further enhance the clarity and translational value of the work.

Round 2

Reviewer 1 Report

Comments and Suggestions for Authors

Congratulations to the authors for the revised version of the manuscript.